# The Enhanced Performance of N-Modified Activated Carbon Promoted with Ce in Selective Catalytic Reduction of NOx with NH₃

**Marwa Saad** *[ID], **Agnieszka Szymaszek** [ID], **Anna Białas** [ID], **Bogdan Samojeden** *[ID] **and Monika Motak** [ID]

Faculty of Energy and Fuels, AGH University of Science and Technology, Al. Mickiewicza 30, 30-059 Krakow, Poland; agnszym@agh.edu.pl (A.S.); anbialas@agh.edu.pl (A.B.); motakm@agh.edu.pl (M.M.)
* Correspondence: marwa@agh.edu.pl (M.S.); bsamo1@agh.edu.pl (B.S.); Tel.: +48-126172119 (B.S.)

**Abstract:** The goal of the study was to modify activated carbon (AC) with nitrogen groups and ceria and to test the obtained materials in low temperature selective catalytic reduction of nitrogen oxides. For that purpose, the starting AC was oxidized with $HNO_3$ of various concentrations, modified with urea and doped with 0.5 wt.% of Ce. It was observed that the increased concentration of acid influenced the catalytic activity, since textural and surface chemistry of activated carbon was changed. The most active sample was that modified with 14 M $HNO_3$ and it reached 96% of NO conversion at 300 °C. Additionally, the addition of Ce improved the catalytic performance of modified AC, and NO was reduced according to oxidation–reduction mechanism, characteristic for supported metal oxides. Nevertheless, the samples promoted with Ce emitted significantly higher amount of $CO_2$ comparing to the non-promoted ones.

**Keywords:** activated carbon; SCR; ceria

## 1. Introduction

Nitrogen oxides ($NO_x$) emitted from the stationary sources are one of the major pollutants introduced into the atmosphere [1–3]. Selective catalytic reduction of $NO_x$ with ammonia ($NH_3$-SCR) is an effective method for elimination of nitrogen oxides from flue gas [4–6]. The commercial catalyst $V_2O_5$-$TiO_2$ promoted with $WO_3$ or $MoO_3$ exhibits satisfactory activity, however it is not free of some disadvantages. The most significant is a narrow temperature window that forces to install the catalyst at high dust position, before electrostatic precipitator and desulfurization installation. In such a position, the catalytic system is exposed to high concentration of $SO_2$ and fly ash that poison the active sites. Additionally, these contaminations can form deposits that plug the catalysts channels [2,7,8]. Moreover, vanadium and tungsten are dangerous to the environment. Thus, safe utilization of the spent catalyst is complicated and relatively expensive [9]. According to the above, there is a high need to develop a novel catalyst that would be active at the low-temperature range in order to enable its placement in the tail-end position. It would allow to avoid the necessity of re-heating of the flue gas. Additionally, in this location the catalytic bed is not vulnerable to the contamination and deactivation by poisons [10].

Recently, activated carbon (AC) was considered as an appropriate material for many applications, such as cleanup processes or catalysis, due to its multifunctional character [11–13]. Extremely high specific surface area, well-developed pore system and relatively low price of the material make it an excellent candidate for the support for substitutive catalyst of $NH_3$-SCR [14–16]. It was reported that modification of activated carbon by the introduction of surface groups with nitrogen and oxygen

improves its catalytic performance in the reduction of NO [4,17]. Li et al. [18] modified low-rank activated coke using a solution of $NH_3 \cdot H_2O$ or $HNO_3$ in order to increase the content of N-containing functional groups and elevate the efficiency of $NH_3$-SCR. X-ray photoelectron spectroscopy (XPS) analysis showed significant changes in the content of N-containing groups before and after catalytic tests, indicating the participation of active pyridine and pyrrole groups and quaternary nitrogen in the surface reaction. On the other hand, the authors reported that the performed modifications led to the formation of non-active groups that can promote or inhibit NO conversion. Moreover, the active groups were easily reduced or oxidized to nitro, nitrate, amine or imine moieties that facilitate side reactions of $NH_3$-SCR. The promotional influence of the introduction of nitrogen onto the surface of activated carbon was also reported by Yao et al. [11]. In agreement with the outcomes presented by Li et al. [18], the researchers postulated that N-doping results in the deposition of pyridynic and/or pyrrolic groups and quaternary nitrogen. The authors found that these structures supply the catalyst with unpaired electrons that are critical for the mechanism of $NH_3$-SCR that assumes adsorption and oxidation of NO on the surface. Additionally, the performed modifications resulted in the increased number of oxygen vacancies that facilitate oxidation of coordinated $NH_3$ to $-NH_2$ intermediates that have positive contribution to low temperature $NH_3$-SCR. Besides, Samojeden and Grzybek [17] reported that promotion with N-groups and modification with transition metals increased the stability of activated carbon-based catalysts at the low-temperature region of $NH_3$-SCR and resistance to deactivation by $H_2O$ present in the gas mixture.

During the past few years, ceria-based catalysts have been examined for the application in $NH_3$-SCR. Although pure $CeO_2$ exhibits very low catalytic activity in $NH_3$-SCR, when used as mechanical or chemical promoter, it can greatly enhance the catalytic properties in $DeNO_x$ process [19–21]. What is more, some studies confirmed that $CeO_2$ can act as an active phase of catalysts for NO abatement itself [22,23]. One of the most important properties of $CeO_2$ is its elevated oxygen transport capacity coupled with the ability to shift between oxidized and reduced states ($Ce^{4+} \rightarrow Ce^{3+}$). Thus, the best recognized promoting effect of ceria on $NH_3$-SCR catalysts is its ability to absorb $O_2$ from the gas phase and form surface oxygen vacancies. These moieties tend to adsorb NO that are transformed to $N_2O_2$ dimers which give rise to $N_2$ upon dissociation [24–26]. Liu et al. [26] analyzed the redox cycle of $V_2O_5$-$WO_3$-$TiO_2$ promoted with $CeO_2$ and assumed that the co-presence of redox couples $V^{5+}/V^{4+}$ and $Ce^{4+}/Ce^{3+}$ accounted for the excellent SCR performance. XPS results showed that the existence of $Ce^{3+}$ could create charge imbalance, vacancies and unsaturated chemical bonds on the catalyst surface which led to more surface chemisorbed oxygen that enhances the catalytic activity at low temperature. Moreover, diffuse reflectance infrared Fourier transform spectroscopy (DRIFTS) measurements revealed that $CeO_2$ contributed to the formation of $NO_2$ and monodentate nitrate species that are reactive intermediates for $NH_3$-SCR. A similar result was observed by Xu et al. [27] who investigated the effect of Ce loading on the performance of V-based catalyst. When Ce content increased, the NO conversion of the promoted catalysts was improved due to the accelerating effect of ceria on the oxidation of $V^{4+}$ and improved ability to absorb, store and transfer the active surface oxygen. Additionally, it was found that the addition of ceria improved the resistance of the catalyst to the presence of $H_2O$ and could expand the temperature window of the commercial $V_2O_5$-$TiO_2$ [28]. Taking into consideration that Ce-containing catalysts exhibit satisfactory activity in NO abatement, we decided to investigate the influence of combined introduction of N-containing groups and ceria on activated carbon and test the catalysts in $NH_3$-SCR.

## 2. Results and Discussion

### 2.1. Nitrogen Adsorption at $-196\,°C$

The results of $N_2$ adsorption at $-196\,°C$ obtained for the analyzed samples are presented in Figure 1. A sharp rise in the $p/p_o$ range of around 0–0.2, and a plateau formed above 0.2 suggests that the materials exhibit isotherm type I, according to IUPAC classification. It indicates microporous

nature of activated carbon. Nevertheless, the presence of hysteresis loop suggests that accurately, the isotherm is a combination of type I and IV of the IUPAC classification [29]. Therefore, the materials also exhibit mesoporosity that has influence on the obtained isotherm profile. The obtained results are in agreement with the scientific literature [30–32]. The structural and textural parameters of the studied samples are presented in Table 1.

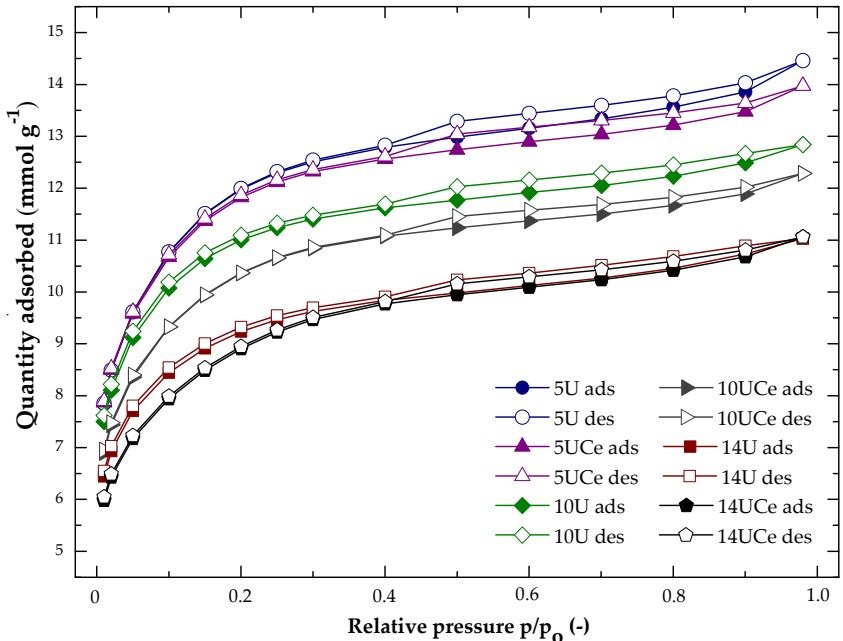

**Figure 1.** The results of $N_2$ adsorption at −196 °C of the analyzed materials.

**Table 1.** Structural and textural parameters of the raw activated carbon and the obtained catalysts.

| No. | Sample Code | $S_{BET}$ $(m^2 \cdot g^{-1})$ | Volume of Micropores $(cm^3 \cdot g^{-1})$ | Average Pore Diameter (nm) | Total Pore Volume $(cm^3 \cdot g^{-1})$ | Micropore Contribution (%) |
|---|---|---|---|---|---|---|
| 1 | AC | 722 | 0.36 | 2.00 | 0.45 | 80.00% |
| 2 | 5U | 813 | 0.47 | 2.36 | 0.50 | 94.00% |
| 3 | 5UCe | 880 | 0.49 | 2.38 | 0.52 | 94.20% |
| 4 | 10U | 829 | 0.44 | 2.22 | 0.48 | 91.60% |
| 5 | 10UCe | 888 | 0.48 | 2.12 | 0.51 | 94.11% |
| 6 | 14U | 812 | 0.42 | 1.98 | 0.46 | 91.30% |
| 7 | 14UCe | 873 | 0.45 | 1.99 | 0.48 | 93.70% |

The initial activated carbon exhibited both the lowest specific surface area and total pore volume among all analyzed catalysts. After modification with 5 M acid, $S_{BET}$ of AC increased from 722 $m^2 \cdot g^{-1}$ to 813 $m^2 \cdot g^{-1}$ and the total pore volume from 0.45 $cm^3 \cdot g^{-1}$ to 0.50 $cm^3 \cdot g^{-1}$. Therefore, the oxidation procedure improved structural and textural properties of AC. As it is presented in Table 1, total pore volume decreased with the increasing concentration of acid used for modification. Similar results were obtained by Vivo–Viches et al. [31]. The authors oxidized activated carbon using $HNO_3$ and observed that the pore volume of the material significantly decreased. Hence, it was suggested that treatment with highly concentrated $HNO_3$ led to gasification and further mechanical destruction of pores by the surface tension of the oxidizing agent. Furthermore, according to the authors, gasification of the mesopore walls led to the blockage of micropores by the oxygenated surface groups. Thus, due to the fact that the total pore volume decreased with the increasing concentration of the oxidizing agent, probably micropores were partly blocked by the oxygenated surface groups. The increase in the specific surface area is very similar in case of all of the modified samples. Thus, it can be assumed that it is independent of the concentration of the used oxidizing agent. Nevertheless, when the concentration of acid was higher, pore volume progressively decreased reaching 0.42 $cm^3 \cdot g^{-1}$ for the modification with 14 M $HNO_3$. Hence, the physical and chemical structure of AC was partially damaged and due to that,

the structure of the pores could collapse, and a certain part of porosity was destroyed. The obtained results are in agreement with those obtained by Lu et al. [33]. Nevertheless, for the considered range of acid concentration, the pore volume of AC did not reach its initial level. The average pore diameter was observed to decrease linearly with the increasing concentration of acid. Therefore, according to the obtained results, the higher concentration of the oxidizing agent, the narrower micropores are formed. It could be the effect of the deposition of urea that covered the pores, causing slight clogging of their inlets [14]. However, in case of 5U the contribution of the micropores was the highest. Probably it was the effect of oxidation of AC in milder conditions in comparison to the those applied for 14U.

The introduction of Ce resulted in the increase of the specific surface area and the volume of micropores, regardless of the applied concentration of the oxidizing agent. In fact, the result is not in agreement with most of the studies over Ce-doped catalysts that indicate the decrease in $S_{BET}$ after introduction of less than 1 wt.% of ceria due to partial porosity blockage [27,34,35]. Nevertheless, since ceria is a well-recognized combustion catalyst that is able to transfer active oxygen from the metal oxide to the carbon surface, it could cause partial gasification of the support. Zou et al. [36] investigated the application of $CeO_2$-$Fe_2O_3$ as the catalyst for biomass gasification and found that the presence of ceria improved oxidative abilities of iron oxide-supported catalyst. Additionally, due to the presence of $CeO_2$, the catalyst was in favor of the oxidation of trace amount of carbon deposits on the catalyst surface. This kind of effect is ascribed to the capacity and mobility of oxygen in $CeO_2$. Depending on various conditions of the reaction, cerium is able to switch between $Ce^{4+}$ and $Ce^{3+}$ and incorporate more or less oxygen in its structure [37]. According to this, the catalysts containing ceria are those among the most effective for soot combustion [38,39]. While $CeO_2$ exchanges its oxygen with gas-phase $O_2$, highly reactive, so-called "active oxygen" species are created. Therefore, it can be assumed that the development of specific surface area of Ce-promoted modified activated carbon was due to the occurrence of the reactions described by Equations (1)–(4) [37]:

$$CeO_2 + C \rightarrow CeO_{2-\gamma} + SOC \text{ (surface oxygen carbon complexes)} \tag{1}$$

$$CeO_{2-\gamma} + \gamma/2\,O_2 \leftrightarrow CeO_2 \tag{2}$$

$$SOC \rightarrow CO/CO_2 + C_f \text{ (free carbon site)} \tag{3}$$

$$\text{O-containing gas} + C_f \text{ (free carbon site)} \rightarrow SOC \tag{4}$$

Probably, the escape of $CO/CO_2$ produced according the reaction described by Equation (3) could be the reason for the development of the specific surface area after introduction of $CeO_2$ on the surface of activated carbon.

Additionally, it is possible that during calcination of the activated carbon with the active phase deposited from the ceria nitrate salt, some amount of $NO_2$ was produced, and it is known that $NO_2$ is even stronger oxidizer than $O_2$ and it could initiate partial combustion of activated carbon. Vratny et al. [40] found that the total decomposition of cerium (III) nitrate hydrate occurs rapidly at about 280 °C. As the calcination temperature was set for 250 °C, the stable $CeO_2$ phase could take part in the chain of the reactions with nitrogen oxides produced upon decomposition of nitrate salt Equations (5)–(9) [37]:

$$CeO_2 + \gamma NO \leftrightarrow CeO_{2-\gamma} + \gamma NO_2 \tag{5}$$

$$CeO_{2-\gamma} + \gamma/2\,O_2 \leftrightarrow CeO_2 \tag{6}$$

$$NO_2 + C \leftrightarrow NO + SOC \tag{7}$$

$$SOC \rightarrow CO/CO_2 + C_f \tag{8}$$

$$\text{O-containing gas} + C_f \rightarrow SOC \tag{9}$$

Thus, similarly to the mechanism containing active oxygen, $NO_2$-associated mechanism of the partial combustion of carbon could be the explanation of the development of the specific surface

area by to the escape of $CO/CO_2$ and penetration of the catalyst pores. Due to the fact that FT-IR studies indicated that N-containing groups introduced onto the surface of activated carbon remained unchanged after introduction of $CeO_2$, it can be concluded that no significant changes of these groups occurred upon the deposition of the active phase.

## 2.2. X-ray Diffraction Analysis

The structural properties of the starting and modified activated carbon samples were examined using X-ray diffraction method. The obtained results are presented in Figure 2. For all of the analyzed catalysts there are two characteristic diffraction maxima at 2θ of ca. 25.0° and 42.9° that correspond to (002) and (100) planes, respectively. The presence of these maxima confirms the existence of graphene sheets and a disordered nature of activated carbon [41,42]. Additionally, with the increasing concentration of the oxidizing agent, intensity of the diffraction maxima decreased. Thus, the modifications led to the disorder of microcrystalline graphite structure and formation of amorphous phases, which is in agreement with the outcomes of $N_2$ adsorption at −196 °C.

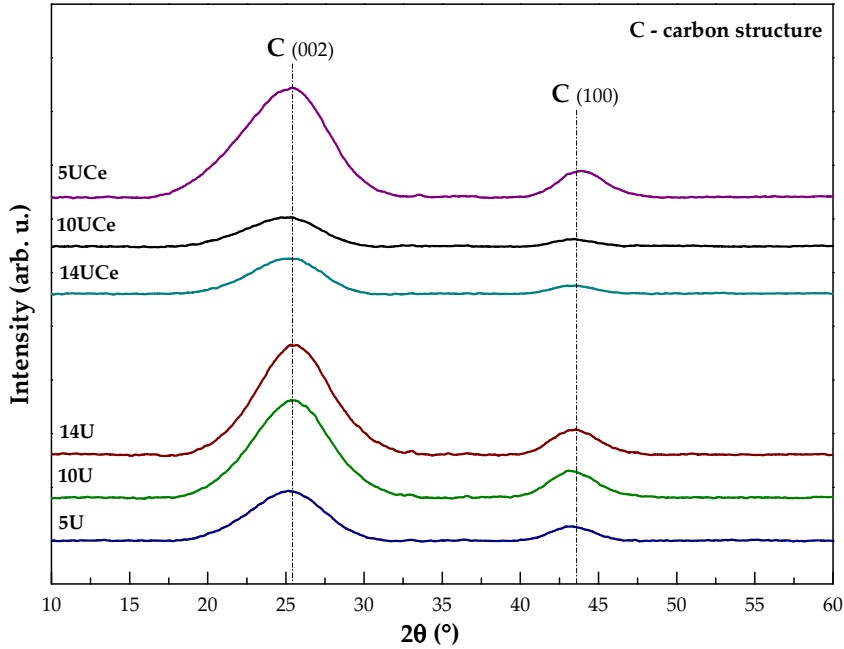

**Figure 2.** XRD patterns of activated carbon modified with N-groups with or without ceria.

No ceria was detected by XRD, due, either, to its low content and/or non-crystalline structure. However, the existence of the additional well dispersed Ce active phase in porous system cannot be excluded. The intensity of the maxima attributed to graphitic sheets significantly decreased. What is more, the width of the maxima increased, suggesting that the addition of ceria lowers the size of AC crystallites and the weight percentage of carbon in the samples. This result is supported by EDX analysis which confirmed the reduction of C percentage in the samples doped with Ce. The most probable reason of the weight loss of C is the destruction of some amount of the organic matter upon calcination after impregnation with the precursor of cerium.

## 2.3. Fourier Transform Infrared Spectroscopy Analysis

The presence of the characteristic functional groups in activated carbon-supported catalysts was examined by FT-IR spectroscopy. The obtained results are presented in Figure 3. It can be noticed that the spectra are divided into two characteristic regions. The first band at 3600–3200 cm$^{-1}$ is assigned to ν(O-H) vibrations of surface hydroxyl groups of activated carbon (3440 cm$^{-1}$) [43]. For raw AC, the band is not well-developed which suggests the lack of stable OH groups in the material. Therefore,

the modification with nitric acid and urea results in the formation of more stable hydroxyl groups on the surface of activated carbon. The second band in range of 1800–1400 cm$^{-1}$ corresponds to $\nu$(C-H) stretching vibrations of carboxylic and 2-pyrone groups (1710 cm$^{-1}$ and 1560 cm$^{-1}$), $\nu$(C=O) and skeletal $\nu_s$(C=C) attributed to aromatic rings (1536 cm$^{-1}$) [43]. More intense peaks detected for the modified samples suggest that oxidation resulted in the increased content of carboxyl and oxygen-containing groups in the samples. The fourth broad band around 1200–900 cm$^{-1}$, composed of three peaks that are very close to each other, is assigned to the presence of aliphatic $\nu$(C-N) groups or $\nu$(N-H) bonding (1120 and 1135 cm$^{-1}$) and asymmetric stretch of $\nu$(C-N-C) (1155 cm$^{-1}$) [44].

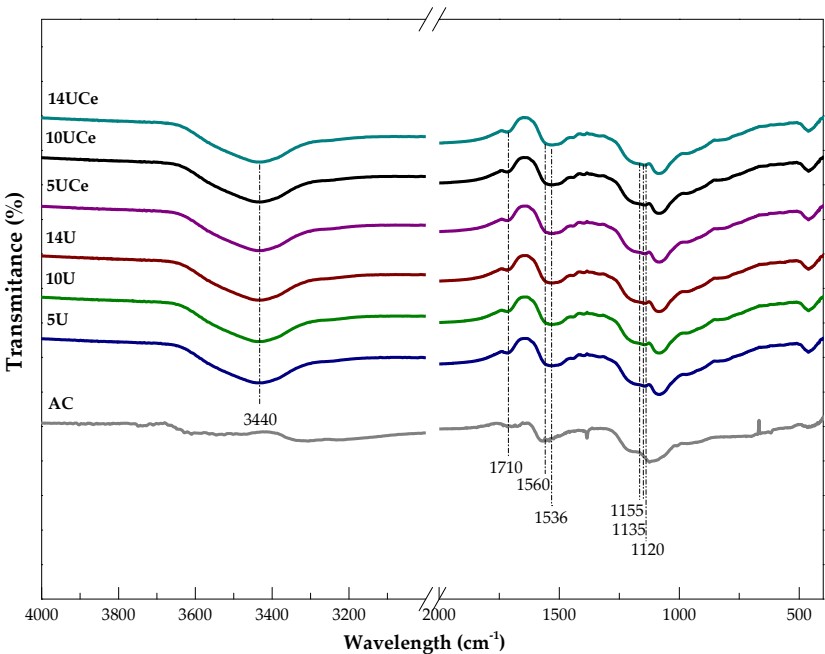

**Figure 3.** FT-IR spectra of activated carbon modified with N-groups with or without ceria.

## 2.4. SEM and EDX Analysis

### 2.4.1. SEM Analysis

SEM and EDX analysis were performed to determine the changes of the surface morphology and amounts of surface elements of the samples modified with nitrogen groups and doped with Ce. Figure 4 shows SEM images of activated carbon oxidized with 5, 10 or 14 M HNO$_3$. Upon modifications, the crystals of AC became more amorphous. Additionally, oxidation with HNO$_3$ changed textural parameters of the samples which is in agreement with the results of N$_2$ adsorption experiment.

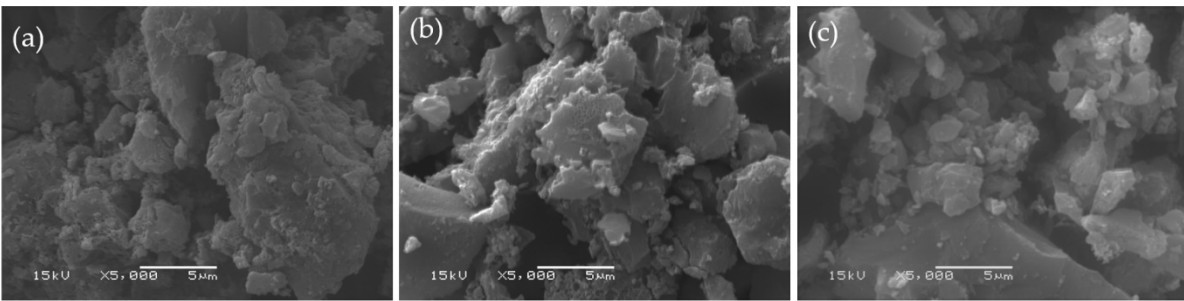

**Figure 4.** SEM pictures of 5U (**a**), 10U (**b**) and 14U (**c**).

In Figure 5a,b, SEM pictures of 14UCe at magnification of 5000 and 20,000, respectively, are presented as an example of the material modified with ceria.

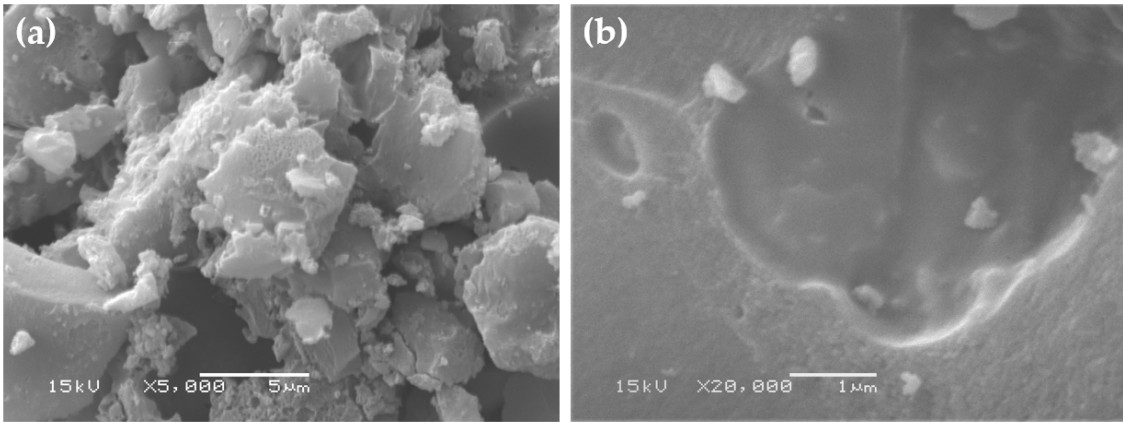

**Figure 5.** SEM pictures of 14UCe at magnification of 5000 (**a**) and 20,000 (**b**).

2.4.2. EDX Analysis

The results of EDX analysis are presented in Table 2. It was observed that with the increasing concentration of $HNO_3$, the amount of O and N was elevated. The opposite effect was noticed for carbon content, suggesting a slight decomposition of organic matter in the presence of more concentrated oxidizer. The introduction of Ce resulted in a higher content of oxygen in comparison to the initial materials. The effect can be explained by the formation of $CeO_2$ on the surface of activated carbon and its ability to capture oxygen [25]. Nevertheless, impregnation with cerium nitrate resulted in a slight decrease in the amount of carbon, probably due to the decomposition of the structure of AC upon calcination procedure. Additionally, Ce-doped samples contained 5–13% less nitrogen in comparison to the initial materials. Possibly, the introduced surface nitrogen groups could interact with ceria and reconstruct or decompose under the influence of surface oxygen supplied by $CeO_2$. Additionally, after introduction of Ce, the impurities of Si traces disappeared. According to our knowledge and literature reports, there could be sorption of ceria on silicon-containing centers that became covered after deposition of $CeO_2$ [45].

**Table 2.** Results of elemental point EDX analysis of the catalysts.

| No. | Sample | Approximate Weight (wt.%) | | | | |
|---|---|---|---|---|---|---|
| | | C | N | O | Ce | Si |
| 1 | 5U | 45.00 | 41.00 | 11.45 | 0 | 2.55 |
| 2 | 5UCe | 41.14 | 36.09 | 20.09 | 2.68 | 0 |
| 3 | 10U | 43.02 | 41.31 | 12.44 | 0 | 3.24 |
| 4 | 10UCe | 40.15 | 35.72 | 21.17 | 2.95 | 0 |
| 5 | 14U | 39.89 | 42.35 | 14.16 | 0 | 3.70 |
| 6 | 14UCe | 38.12 | 40.28 | 21.36 | 0.23 | 0 |

*2.5. Thermogravimetric Analysis*

Thermogravimetric analysis was carried out in order to analyze the stages of decomposition of the AC-based materials in the temperature range of 30–800 °C. The results obtained for 5U, 10U and 14U are depicted in Figure 6a. The mass loss of the samples in the temperature window of $NH_3$-SCR tests are presented in Table 3. First of all, it can be observed that thermal stability of the oxidized activated carbon decreased with the increasing concentration of nitric acid used for the modification. It is caused by the elevated content of oxygen groups deposited due to the interaction with the concentrated acid and as a result, higher reactivity of carbon during thermal decomposition [46].

In general, three stages of decomposition in range of 30–80 °C, 80–380 °C and 380–800 °C, respectively, can be distinguished for the materials without Ce. The first one is attributed to the evaporation of moisture from the materials [47]. The decrease in mass during the second stage can be attributed to the decomposition of some functional groups, such as hydroxylic or carboxylic [48]. A different loss of mass for the materials in this temperature region can be the result of various amount of these groups on the surfaces. Additionally, this temperature range includes the temperature window of $NH_3$-SCR. Therefore, negligible weight loss suggests that the catalysts exhibit satisfactory stability during the reaction. The third stage of the decomposition starts above 380 °C and occurs due to the decomposition of the carbon skeleton [47,48]. What is more, the material treated with 14 M $HNO_3$ exhibited the lowest thermal stability above 500 °C. It suggests that under the influence of highly concentrated oxidizing agent, structural stability of activated carbon decreases. The effect can be explained by the fact that the introduction of higher amount of oxygen and hydrogen onto the surface of carbon increases its reactivity, leading to rapid combustion of carbon [46,49]. TGA profiles obtained for the materials doped with Ce are presented in Figure 6b–d. Thermal stability after introduction of Ce decreased significantly in the order of 14UCe > 10UCe > 5UCe. The presence of Ce shifted the initial temperature of the third stage of decomposition from around 380 °C for all non-promoted samples to about 280 °C for 5UCe and 300 °C for both 10UCe and 14UCe. What is more, there is a noticeable difference between the weight loss of the samples non-doped and doped with Ce during the second stage of decomposition, and it progressively increases with the increasing temperature. The phenomenon can be caused by the evaporation of $H_2O$ and $CO_2$ during the first stage of decomposition and further formation of cerium carbonate or cerium oxycarbonate hydrate $Ce_2O(CO_3)_2 \cdot H_2O$ due to the interaction of the gases with dispersed $Ce^{3+}$ species. Liu et al. [50] reported that the formation of cerium oxycarbonate hydrate can occur according to the reactions described by Equations (10)–(13):

$$H_2NCONH_2 + H_2O \rightarrow 2NH_3 + CO_2 \tag{10}$$

$$NH_3 + H_2O \rightarrow NH_4^+ + OH^- \tag{11}$$

$$CO_2 + H_2O \rightarrow CO_3^{2-} + 2H^+ \tag{12}$$

$$2Ce^{3+} + 2OH^- + 2CO_3^{2-} \rightarrow Ce_2O(CO_3)_2 \cdot H_2O \tag{13}$$

The reaction described by Equation (10) involves urea as a reactant. In fact, urea used for the introduction of the N-groups onto the surface of activated carbon was fully decomposed upon calcination procedure. Nevertheless, the nitrogen groups present in activated carbon could decompose during the analysis and except of $CO_2$ and $H_2O$, ammonia could have also been formed. Subsequently, carbonate ions formed according to Equation (12), hydroxyl anions and $Ce^{3+}$ species present on the surface of AC underwent surface reaction forming $Ce_2O(CO_3)_2 \cdot H_2O$. The fast rise of cerium oxycarbonate hydrate upon the cited reactions below 100 °C was also confirmed by the studies carried out by Ikuma et al. [51]. The postulates of Liu et al. [50] and Ikuma et al. [51] are in agreement with the obtained outcomes, since the temperature of $Ce_2O(CO_3)_2 \cdot H_2O$ formation fits the first temperature range of TGA experiment. Furthermore, thermal stability analysis carried out by Liu et al. [50] confirmed that decomposition of cerium oxycarbonate hydrate occurs around 250–450 °C. Thus, it can be concluded that $Ce_2O(CO_3)_2 \cdot H_2O$ was formed not only on the surface of 5UCe, but also on 10UCe and 14UCe. Due to that, the temperature of decomposition was shifted to around 280–300 °C and the reaction described by Equation (14) took place [50]:

$$2Ce_2O(CO_3)_2 \cdot H_2O + O_2 \rightarrow 4CeO_2 + 2H_2 \tag{14}$$

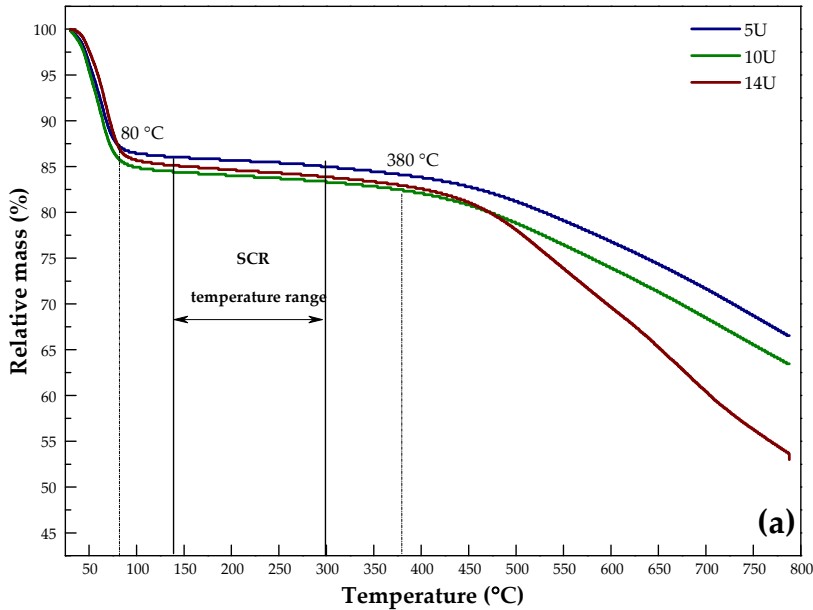

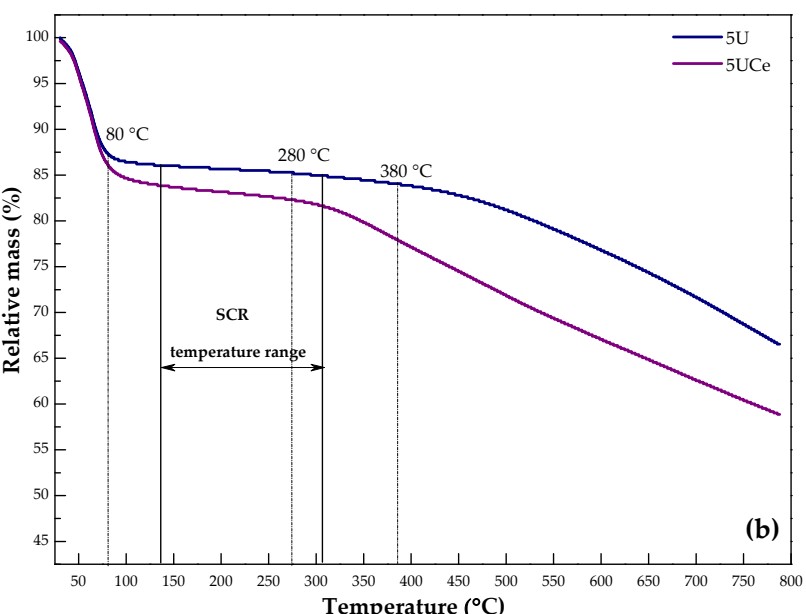

**Figure 6.** *Cont.*

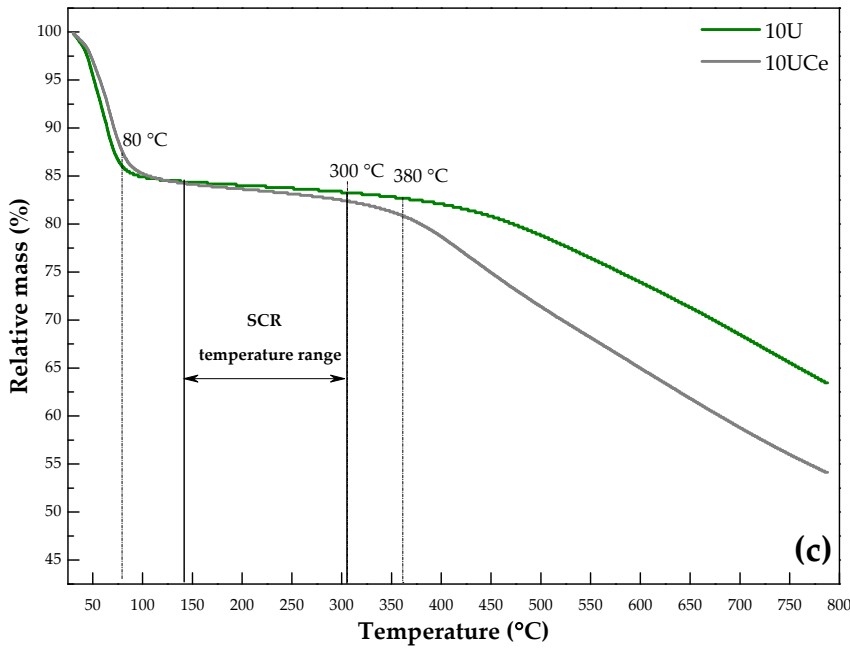

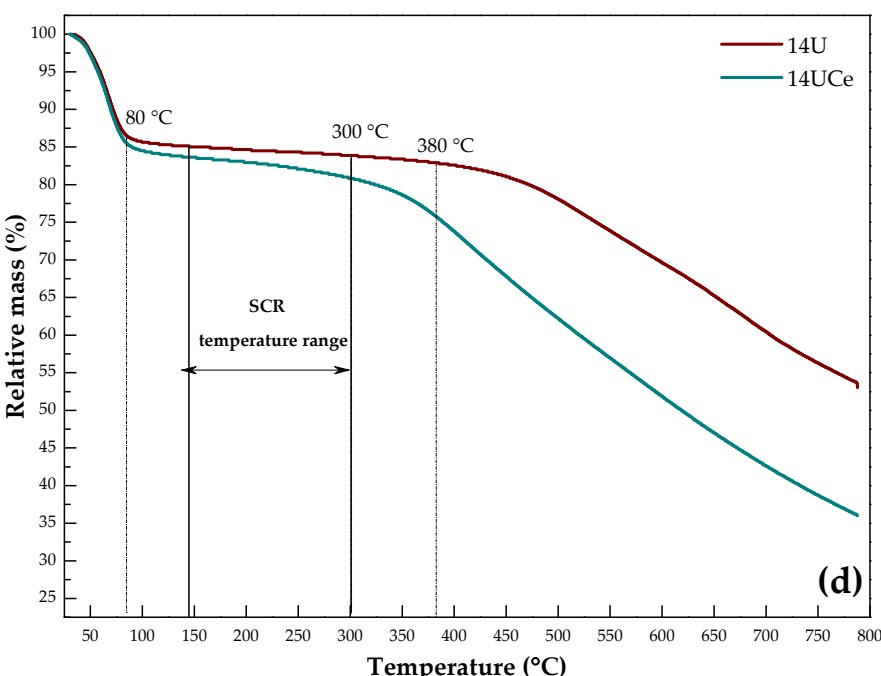

**Figure 6.** TGA analysis of activated carbon modified with nitric acid of different concentrations (**a**) and comparison of the results of TGA analysis obtained for 5U and 5UCe (**b**), 10U and 10UCe (**c**), 14U and 14UCe (**d**).

**Table 3.** Weight loss of the samples in the second temperature range of TGA experiment.

| Sample | Weight Loss in the Particular Temperature Range (%) | |
|---|---|---|
| | 80–380 °C | 140–300 °C |
| 5U | 3.03 | 1.06 |
| 10U | 3.27 | 1.06 |
| 14U | 3.48 | 1.12 |
| | 80–300 °C | 140–300 °C |
| 5UCe | 1.87 | 1.50 |
| 10UCe | 3.42 | 1.73 |
| 14UCe | 4.26 | 3.04 |

Alternatively, the higher mass loss in case of the materials doped with Ce can be explained by the ability of $CeO_2$ to store and release oxygen [20]. Due to that, $CeO_2$-containing catalysts could be continuously oxidized during TGA. Additionally, ceria is recognized as an active combustion catalyst [37,38] and led to the partial combustion of activated carbon at temperatures higher than 380 °C. What is more, loading of metal oxide could lead to the reconstruction of functional groups introduced upon modifications with nitric acid and urea. As a result, thermal stability of the material can decrease which was confirmed by the findings of Lu et al. [33] who modified activated carbon with nitrogen groups and high amount of ceria and drew similar conclusions. Additionally, since reactivity of carbon is strongly correlated with hydrogen and oxygen content [52], high amount of these elements caused by the performed modifications could increase its tendency to undergo oxidation reactions and combustion of carbon above 380 °C [46,49].

### 2.6. Catalytic Performance Tests

The catalytic activity of the samples modified with $HNO_3$ and ceria was measured in the temperature range of 140–300 °C, which stands for the low-temperature range of $NH_3$-SCR. The results of catalytic tests are presented in Figures 7–9. Due to the fact that $CO_2$ emission at particular temperature ramps increased from less than 50 ppm below 260 °C to more than 3000 ppm at 300 °C, the results were presented in two separated figures (Figure 9a,b), in order to show the outcomes in the most visible way.

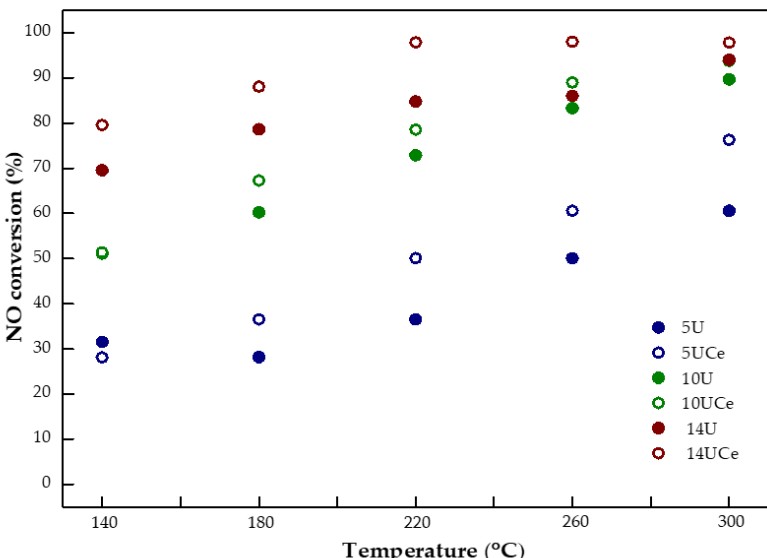

**Figure 7.** Results of NO conversion obtained for the catalysts.

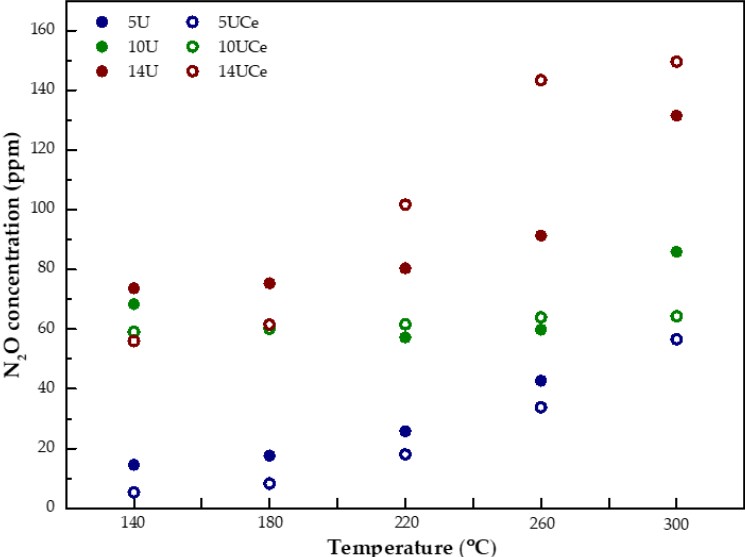

**Figure 8.** Results of N$_2$O concentration in the exhaust gas obtained for the catalysts.

It can be observed that modification with more concentrated HNO$_3$ improved catalytic performance of modified activated carbon in NO reduction. The highest conversion was obtained for 14U, since at 140 °C about 69% of NO was reduced. Additionally, temperature of 50% conversion (T$_{50}$) for that sample was below the analyzed temperature range, while for 5U and 10U it was 260 °C and 140 °C, respectively. The highest activity of 14U can be explained by the abundance in the introduced functional groups onto the surface of AC. Grzybek et al. [14] compared the catalytic activity of activated carbon promoted and non-promoted with N-groups in NH$_3$-SCR. The authors concluded that oxidation with HNO$_3$ followed by treatment with urea or ammonia was a key factor in the improved catalytic performance. XPS studies carried out by the researchers indicated that pyridynic or pyrrole/pyridine groups were formed upon the treatment with HNO$_3$ and urea or ammonia. These groups were found to be essential to facilitate the adsorption of NO from the gas phase and accelerate the SCR process according to the Langmuir–Hinshelwood mechanism [53–55]. Furthermore, Lin et al. [54] postulated that after modification with urea, the phenolic hydroxyl oxygen of non-modified activated carbon is transformed to carbonyl oxygen of quinine and pyridine, pyrrole and quaternary nitrogen functional groups. Additionally, the authors investigated the behavior of activated carbon modified with N-groups during SCR reaction using in situ DRIFTS analysis. The results indicated that pyridine, pyrrole and quaternary amine groups were the centers for NO adsorption while the phenolic hydroxyl group was an adsorption site for NH$_3$. Thus, it is predicted that in case of the analyzed samples, pre-treatment with the most concentrated acid introduced the highest amount of those groups. In reference to elemental point EDX results, the amount of surface oxygen and nitrogen could be slightly increased for 14U in comparison to 5U and 10U. Since surface active oxygen plays an important role in the mechanism of NH$_3$-SCR [1], it can be predicted that its higher concentration could be the reason of better catalytic activity of 14U. Therefore, the number of the groups depends additionally on the concentration of the oxidant. Nonetheless, Samojeden et al. [4], who investigated the influence of the modification of activated carbon on its performance in NH$_3$-SCR, postulated that the matter of oxidation is complicated by the fact that both the initial structure of carbonaceous materials and the type of oxidative pretreatment play an important role and only semi-quantitative prediction on the type and number of groups may be generally given. What is more, since textural properties of the catalysts, confirmed by BET analysis, are similar regardless of the concentration of HNO$_3$, the highest activity is possibly correlated with the formation of the functional groups that are the most active in NH$_3$-SCR. The sample 14U emitted the highest amount of N$_2$O among all of the analyzed materials and the obtained effect is in a good agreement with the literature [4,56,57]. Li et al. [18] suggested that N-groups can undergo regular changes during SCR reaction, such as reduction by NH$_3$ or oxidation by

$O_2$ present in the gas stream. The resulting nitro, nitrate, amine and imine groups were confirmed by the authors to decrease selectivity to $N_2$ and formation of by-products, such as $N_2O$. The emission of $CO_2$ was similar for 5U and 10U in the temperature range of 140–220 °C, but with the increasing temperature it significantly increased for 10U. Moreover, the highest amount of $CO_2$ was emitted by 14U. For this sample, at 300 °C the concentration of carbon dioxide reached ca. 2400 ppm which can be the reason of pre-treatment with strongly oxidizing agent and faster decomposition of organic matter, confirmed by the results of TGA. Possibly, the highest amount of oxygen groups introduced on the surface of AC lowered the strength of the structure of activated carbon and increased its tendency to decomposition.

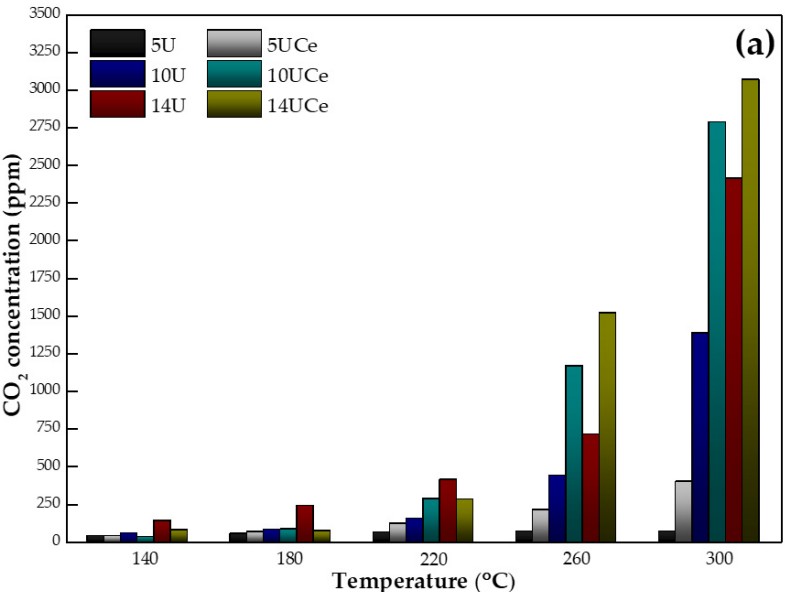

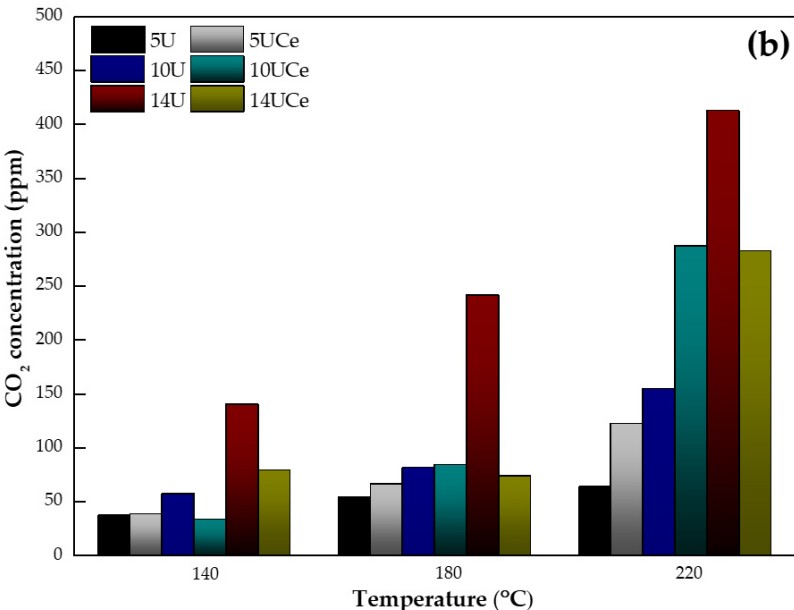

**Figure 9.** Results of $CO_2$ concentration in the exhaust gas obtained for the catalysts: (**a**) at all temperature ramps of the reaction and (**b**) at temperature ramps below 260 °C.

While comparing the catalytic performance of the samples with and without Ce, it can be observed that the addition of ceria improved catalytic activity of modified activated carbon to some extent. The promoting effect was rather independent of the concentration of $HNO_3$ used for the modification procedure. Marberger et al. [58] suggested that in the lower temperature range of SCR, Brönsted acid sites are not essential for the activity of the catalyst, since the catalytic performance depends on the adsorption of NO and its interaction with gas-phase $O_2$. On the catalysts doped with metal oxides reduction of the adsorbed NO by $NH_3$ is possibly due to electron transfer that affects the generation of oxygen vacancy and improves the catalytic performance [33,59,60]. EDX analysis confirmed that the amount of oxygen is significantly higher for the samples doped with $CeO_2$, thus surface adsorbed oxygen species could increase the activity of oxidation–reduction mechanism of SCR that was proposed by Shen et al. [22]. According to the authors, within the reaction, the reduced metal oxide is recovered to its initial state due to its interaction with surface adsorbed oxygen. Therefore, its higher amount accelerates regeneration of the active site. Furthermore, it was confirmed that $CeO_2$ undergoes an oxidation–reduction process according to reactions described by schematic cycle presented in Figure 10.

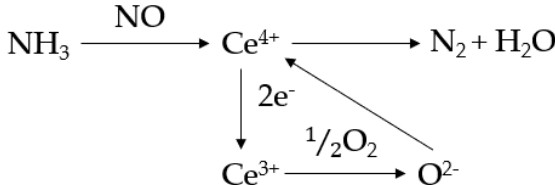

**Figure 10.** Oxidation–reduction process of $NH_3$-SCR that involves ceria active sites [21].

Nitrogen oxide adsorbed on $Ce^{4+}$ forms chemically adsorbed species that subsequently adsorb $NH_3$, react with lattice oxygen of $Ce^{4+}$ and yield nitrogen and water vapor. The compensation of the lack of oxygen of $Ce^{4+}$ occurs upon its transfer from the adjacent $Ce^{3+}$. Additionally, oxygen from the gas phase can be adsorbed on $Ce^{3+}$, transformed into lattice oxygen and transferred to $Ce^{4+}$. It enables fast regeneration of the reduced metal active site and proceeding of oxidation–reduction mechanism. On the other hand, some researchers confirmed that on AC modified with $HNO_3$ reduction of NO proceeds according to the mechanism that involves initial oxidation of NO [12,33,61,62]. Thus, the difference in catalytic performance in doped and non-doped samples might originate from different mechanisms that take place on the studied catalysts.

The amount of $CO_2$ emitted during particular temperature ramp of the catalytic test was calculated as the average concentration in the ramp. It was found that concentration of the gas increased considerably after introduction of Ce, regardless of the concentration of the acid used for modification. The highest difference in carbon dioxide emission can be observed above 260 °C, in case of all of the samples. Since the results of TGA analysis confirmed that the introduction of ceria resulted in the more rapid degradation of the carbonic matter, the emission of $CO_2$ is probably caused by the proceeding degradation of the catalyst with the increasing temperature. The effect is especially visible for 14UCe. Hence, acid treatment combined with the modification with $CeO_2$ decreased stability of the catalysts. An alternative explanation of the phenomenon might be a direct involvement of the support in the catalytic reaction. Lu et al. [33] found that carbon from the support can take part in NO reduction and behave as either a catalytic center or a reducing agent. The authors suggested that $CO_2$ is emitted upon this process according to the reaction described by Equation (15):

$$2NO \ + \ C \ \rightarrow \ N_2 + CO_2 \qquad (15)$$

Additionally, $CO_2$ could be also formed by the interaction of active phase with the support, that finally resulted in the reduction of NO by CO described by Equations (16)–(18):

$$CeO_2 \ + \ AC \ \rightarrow CeO + CO \qquad (16)$$

$$CeO_2 + CO \rightarrow CeO + CO_2 \tag{17}$$

$$2CO + 2NO + O_2 \rightarrow N_2 + 2CO_2 \tag{18}$$

However, in comparison to the amount of $CO_2$ produced due to the decomposition of the organic matter of AC, its concentration generated according to reactions described by Equations (17) and (18) is minor.

The amount of $N_2O$ did not change considerably after the promotion with Ce in case of 5U and 10U. However, for 14U and 14UCe, the difference between the amount of $N_2O$ emitted during the reaction reached 22 ppm at 220 °C, 53 ppm at 260 °C and 31 ppm at 300 °C. It could have resulted from the interaction of higher amount of nitrogen groups with $CeO_2$, the latter being able to store oxygen and possibly, in some way oxidize N-species on the catalyst surface during the catalytic reaction.

## 3. Materials and Methods

### 3.1. Catalysts Preparation

Activated carbon utilized for the purpose of the studies was provided by Gryfskand (Hajnówka, Poland). The material was oxidized using the solutions of 5 M, 10 M or 14 M nitric acid ($HNO_3$) in the ratio of 15 mL acid/1 g of AC. The samples of activated carbon were left to react with $HNO_3$ at 90 °C for 2 h, filtered, washed with distilled water to reach pH ca. 7.0 and dried at 110 °C overnight.

N-containing groups were incorporated onto the surface of acid-modified AC by incipient wetness impregnation with 5 wt.% aqueous solution of urea. Afterwards, the samples were dried at 110 °C overnight and calcined in the flow of 2.25 vol.% $O_2$ in helium at 350 °C for 2 h. The samples were subsequently ground and sieved to obtain a fraction of 0.25–1 mm. The deposition of 0.5 wt.% of cerium was performed by incipient wetness impregnation with a solution of cerium nitrate. The samples were dried at 110 °C overnight and calcined in a stream of helium at 250 °C for 1 h. The obtained samples are listed in Table 4.

**Table 4.** List of the prepared catalysts (where "+"—sample modified with ceria; "-"—sample non-modified with ceria).

| No. | Sample Code | Concentration of $HNO_3$ Used for The Synthesis ($mol \cdot dm^{-3}$) | Modification with Ce |
|---|---|---|---|
| 1 | AC | non-modified sample | |
| 2 | 5U | 5 | - |
| 3 | 5UCe | 5 | + |
| 4 | 10U | 10 | - |
| 5 | 10UCe | 10 | + |
| 6 | 14U | 14 | - |
| 7 | 14UCe | 14 | + |

### 3.2. Catalysts Characterization

The determination of the specific surface area of the obtained catalysts was performed by nitrogen adsorption at −196 °C using Gemini V2.00 2380 Micrometrics apparatus (Micrometrics, Norcross, GA, USA) at −196 °C. Before the analysis, each sample was outgassed at 150 °C for 2 h. The Brunauer, Emmett and Teller method was used to calculate the specific surface area ($S_{BET}$) of the samples, while the pore distribution was determined by Barrett-Joyner-Halenda (BJH) method. The radius of the pores were determined by the Kelvin–Thomson Equation (19) [63]:

$$\frac{1}{r_{k,x}} - \frac{1}{r_{k,y}} = -\frac{RT}{\sigma V_m} ln \frac{p}{p_o} \tag{19}$$

where $r_{k,x}$, $r_{k,y}$—curvature radius of the meniscus, σ—surface tension of the liquid adsorbate, $V_m$—molar volume of the adsorbate, $R$—gas constant, $T$—absolute temperature, $p$—pressure of the vapor above the meniscus at which condensation or evaporation occurs and $p_o$—saturated vapor pressure above the flat surface of the liquid adsorbate. Total pore volume was obtained from the pore volume distribution curve. It was calculated by the differentiation of the function described by Equation (20) [63]:

$$V = f(r_k) \tag{20}$$

where $V$ is pore volume while $r_k$ is a radius of the pore derived from the Kelvin–Thomson equation, Equation (19). Thus, pore volume distribution curve can be presented by Equation (21) [63]:

$$dV/dr = f(r_k) \tag{21}$$

All of the calculations were made by the software of Gemini V2.00 2380. Structural analysis was carried out using X-ray diffraction technique. The X-ray diffraction patterns were obtained using Empyrean (Panalytical) diffractometer (Panalytical, Almelo, UK). The instrument was equipped with copper-based anode (Cu-Kα LFF HR, λ = 0.154059 nm). The diffractograms were collected in 2θ range of 2.0–80.0° (2θ step scans of 0.02° and a counting time of 1 s per step). Fourier-transform infrared spectroscopy (FT-IR) was used to examine the presence of characteristic chemical groups in activated carbon. The spectra were recorded on a Frontier MIR/FIR Spectrometer in the region of 4000–400 cm$^{-1}$ with a resolution of 4 cm$^{-1}$. Before measurement, each sample was mixed with KBr (the ratio of 1:100) and pressed into disk. Thermogravimetric analysis (TGA) was carried out in the temperature range of 20–800 °C and in a controlled atmosphere of nitrogen (100 cm$^3 \cdot$min$^{-1}$) using the SDT Q600 V20.9 Build 20 instrument (TA Instruments, New Castle, DE, USA). The morphology and chemical composition of the synthesized catalyst were examined by scanning electron microscopy (SEM) that was carried by JEOL JSM 6360LA, (JEOL Ltd., Tokyo, Japan) at a magnification of 5000 and 10,000. Energy-dispersive X-ray spectroscopy (EDX) was carried out by Shimadzu 7000 spectrometer (Shimadzu Corporation, Kyoto, Japan) with an energy dispersive X-ray analyzer. The elemental analysis was performed at 20 kV (beam voltage) in a vacuum chamber. The acquisition time of the spectra was 300 s, the beam current was 86 uA and the detector dead time was of about 30%. The elemental composition of the examined catalysts was analyzed using the ZAF method.

### 3.3. Catalytic Tests

NH$_3$-SCR catalytic tests were carried out in fixed-bed flow microreactor under atmospheric pressure. The reaction mixture (800 ppm of NO, 800 ppm of NH$_3$, 3.5 vol.% of O$_2$) was introduced to the microreactor through mass flow controllers and helium was added as a balance to maintain the total flow rate of 100 cm$^3 \cdot$min$^{-1}$. The concentrations of residual NO, N$_2$O and CO$_2$ were analyzed downstream of the reactor by FT-IR detector (ABB 2000 A0 series) (ABB Automation GmbH Measurement & Analytics, Frankfurt am Main, Germany). The experimental NH$_3$-SCR process was conducted at the temperature ramps of 140, 180, 220, 260 and 300 °C (duration of each ramp was 30 min) and NO conversion was calculated according to the formula represented by Equation (22):

$$NO_{conversion} = \frac{NO_{in} - NO_{out}}{NO_{in}} \tag{22}$$

where $NO_{in}$—inlet concentration of NO and $NO_{out}$—outlet concentration of NO.

## 4. Conclusions

In the presented work activated carbon was oxidized with nitric acid of various concentrations, modified with urea in order to introduce N-functional groups, subsequently promoted with ceria and tested as the catalyst of NH$_3$-SCR. NO conversion increased while AC was pre-treated with more

concentrated $HNO_3$. However, the most active sample emitted the highest amount of $N_2O$ and $CO_2$ during the reaction. The improvement of the catalytic activity with the increasing concentration of nitric acid can be explained by the more disordered graphite structure and/or introduction of higher amount of surface active groups, which is in agreement with the studies presented in scientific literature. These groups were considered to provide additional active centers to adsorb NO and $NH_3$. The introduction of ceria increased NO conversion for all of the catalysts and decreased the concentration of $N_2O$ due to the ability of $CeO_2$ to store oxygen. Nevertheless, because of the formation of deposits, such as cerium oxycarbonate hydrate, and the oxidizing nature of Ce, its introduction decreased thermal stability of activated carbon which led to excessive emission of $CO_2$ during the reaction above 220 °C. Additionally, some amount of $CO_2$ could be produced within SCR process by direct involvement of AC in the reaction and oxidation of CO to $CO_2$.

**Author Contributions:** The experimental work was designed and supported by B.S., M.M., M.S. prepared the catalysts; performed the catalytic experiments, characterization of catalysts, and analysis of data; A.B. supported the catalytic experiments and characterization of catalysts; M.S. writing—Original draft preparation; A.S., writing—Review and editing; B.S., review, supervision; M.M., supervision. All authors have read and agreed to the published version of the manuscript.

**Funding:** This work was funded by Grant AGH 16.16.210.476.

**Conflicts of Interest:** The authors declare no conflict of interest.

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
