# Peer review of "The Enhanced Performance of N-Modified Activated Carbon Promoted with Ce in Selective Catalytic Reduction of NOx with NH3"

_catalysts, doi:10.3390/catal10121423_

Round 1
Reviewer 1 Report
The authors report on the performance of several modified carbon catalysts for the SCR of NOx with NH3. Overall, the modifications do appear to increase catalytic activity, however, there are several issues that are of concern.
Firstly, the figures could be better presented and rather than use closed symbols for both adsorption and desorption or similar colours for figure 2 these could be adjusted to use an open symbol for desorption or label the lines a, b, c, d… in figure 2. Then utilise the caption to describe the contents and the symbols or the corresponding catalysts. Some of the SEM images do not have space bars, please amend these.
Despite what feels like unnecessary repetition in Figs. 9-12 the data as displayed indicates that the experiments were conducted over some undisclosed time. The linked data points with lines of Figs. 9-11 contrast with the bar charts of Fig. 12. The presumption is that the experiments were conducted through ramping the reaction temperature of one catalyst formulation until 300C was reached. Was this the case? If so, how did this differ across the catalysts surveyed in this report? Furthermore, if this is the case then are the catalysts stable at say 250C for the duration of the experiment?
The authors conclude that the highest conversion was over the 14U catalyst (of the non-Ce added catalysts) and this was due to the significant N-containing groups on the material. Furthermore, the contrast of the N- and O- content of the 5U and 10U samples to the 14U one is minor at best and could be within measurement error of EDX. The textural properties of these materials are also comparable and it would appear that understanding the higher activity of 14U would require further analysis and perhaps not just following literature examples. The N-containing groups on the surface may well be different with comparison of 5U and 14U, however, this is not clear.
The high relative release of CO2 from the Ce-containing materials could be explored through weighing the catalyst before and after reaction. Here the graphs (Fig. 12), as presented do not make it clear if this is an average of some time or simply a single point once the reaction temperature was reached.
I can recommend publication with changes to the manuscript to address these issues.
Author Response
The enhanced performance of N-modified activated carbon promoted with Ce in selective catalytic reduction of NOx with NH3
The authors report on the performance of several modified carbon catalysts for the SCR of NOx with NH3. Overall, the modifications do appear to increase catalytic activity, however, there are several issues that are of concern.
Firstly, the figures could be better presented and rather than use closed symbols for both adsorption and desorption or similar colours for figure 2 these could be adjusted to use an open symbol for desorption or label the lines a, b, c, d… in figure 2. Then utilise the caption to describe the contents and the symbols or the corresponding catalysts. Some of the SEM images do not have space bars, please amend these.
Response: Thank you for your valuable comment. According to your suggestion, we applied closed symbols for adsorption and opened for desorption. Since the caption is included in the figure, we avoided additional incorporation of the samples’ names directly next to the plotted data. We hope that right now it is much easier to conclude which symbol corresponds to particular sample. Additionally, as you recommended, we utilized captions in Figure 2. and Figure 3., so that is much easier to correlate appropriate plot to each sample. We also pasted SEM images with their space bars to the manuscript.
Despite what feels like unnecessary repetition in Figs. 9-12 the data as displayed indicates that the experiments were conducted over some undisclosed time. The linked data points with lines of Figs. 9-11 contrast with the bar charts of Fig. 12. The presumption is that the experiments were conducted through ramping the reaction temperature of one catalyst formulation until 300C was reached. Was this the case? If so, how did this differ across the catalysts surveyed in this report? Furthermore, if this is the case then are the catalysts stable at say 250C for the duration of the experiment?
Response: Repetition of the data in Figures 9-12. is very apt comment, thank you. Hence, we presented our results on 3 schemes that represent conversion of NO, and concentration of N2O and CO2 obtained for all of the samples. Your presumptions according to the conduction of catalytic tests through ramping of the reaction temperature is correct, we included deeper description of the catalytic tests in the “Catalytic tests” paragraph. According to that, we decided to change lines in Fig 8. that now represents NO conversion into closed and opened circles to depict activity of the catalysts within particular temperature ramps more clearly. Indeed, taking into consideration the temperature of 250ºC, we can only conclude by the outcomes obtained at 260ºC and 300ºC that the catalysts preserve their stability, since the catalytic activity increases in the temperature range of 240-300ºC.
The authors conclude that the highest conversion was over the 14U catalyst (of the non-Ce added catalysts) and this was due to the significant N-containing groups on the material. Furthermore, the contrast of the N- and O- content of the 5U and 10U samples to the 14U one is minor at best and could be within measurement error of EDX. The textural properties of these materials are also comparable and it would appear that understanding the higher activity of 14U would require further analysis and perhaps not just following literature examples. The N-containing groups on the surface may well be different with comparison of 5U and 14U, however, this is not clear.
Response: Thank you for that valuable remark. You are absolutely right that the minor difference in the amount of N and O detected by EDX can result from the measurement error of the apparatus. Due to that, we included that information in the manuscript and also, admitted that textural properties are not directly responsible for the elevation of catalytic activity of 14U. Since we are not able to confirm directly the type and concentration of functional groups present on the analyzed samples because of very limited period of time given for the corrections of the manuscript, we can only support out conclusions by the findings presented in scientific literature. However, we can observe in FT-IR spectra of the samples that the modified materials contain better-developed peaks that are assigned to nitrogen active groups in comparison to the raw activated carbon. Indeed, we cannot observe significant changes in the spectra for the increasing HNO3, nonetheless, combination of our outcomes with previous studies performed in similar field can partly confirm our suggestions. Thus, in order to clarify the issue, we corrected the part of the manuscript where we admit that higher activity is related to the increased amount of N and O or the higher specific surface area. We hope that right now the part is better to understand and accurate.
The high relative release of CO2 from the Ce-containing materials could be explored through weighing the catalyst before and after reaction. Here the graphs (Fig. 12), as presented do not make it clear if this is an average of some time or simply a single point once the reaction temperature was reached.
Response: Thank you for the valuable comment.
In our research, we use a reactor in which the catalytic bed is placed on a glass wool to prevent small particles from entering the measuring system. Therefore, we do not weigh the spent catalyst, due to possible large differences of weight, because some of the fine particles of the catalyst can stay in the wool (bed).
Nevertheless, we will consider using reactors with a different design to weigh the catalysts after the reaction.
As we mentioned in our earlier explanations, we corrected the figures related to the catalytic tests and we presented more clearly the obtained results of CO2 concentration in the flue gas. Additionally, we mentioned in the manuscript that concentration of CO2 is calculated in the particular temperature range as the average.
I can recommend publication with changes to the manuscript to address these issues.

Reviewer 2 Report
In this manuscript, N and O functionalized activated carbons were prepared by treatment with urea and nitric acid. Moreover, ceria was also introduced into the functionalized activated carbon by incipient wetness impregnation. The catalytic performance of the synthesized activated carbons was tested in the NO reduction in presence of NH3. The samples have been extensively characterized; however, the results are not properly discussed. Thus, I consider that the quality of the manuscript is not enough for its publication in Catalysts and several issues must be improved previous to its further consideration for publication:
Minor revisions:
- English must be checked and corrected. There are some sentences in which the verb is missing, e.g.: abstract: “the goal of the study to modify activated carbon with …”. Pag 1. Line 27: isadvantages must be disadvantages.
- Table 1. The equations and methods used to calculate all the parameters of Table 1 must be defined. Is “Pore volume” the volume of micropores? Is total pore volume calculated as the N2 adsorbed at 0.95 minus the micropores volume?. Has the average pore diameter been calculated from the DFT pore size distribution?
- For a proper characterization of activated carbons, Hg-porosimetry must be also performed in order to characterize de wide mesoporosity and macroporosity.
- Line 98. “Total pore volume from 0.36 cm3 g-1 to 0.50 cm3 g-1”. The total pore volume increases from 0.45 to 0.50 cm3 g-1 when the activated carbon is treated with 5 M acid.
- Line 129-130. “The presence of those maxima confirms the existence of graphene sheets and disordered nature of activated carbon 36”. Is this 36 the reference?.
- Figure 7. Figure caption (b) is missing.
- Line 309. Correct “non-modifiedactivated carbon”
- Equation 6 is not correct. Remove the number 2 of ceria to properly balance the equation
- Which activated carbon was used? Is it a commercial one or was it synthesized in your laboratory?
- Table 2. This table does not reflect the real preparation conditions because samples with U (5U, 10 U and 14 U) were also modified with urea, whereas the samples with Ce were treated with the same procedure described for U samples but Ce was also introduced. Please modify this table in order to avoid misunderstanding.
- It is better use “nitrogen adsorption at -196 ºC” rather than “low-temperature nitrogen sorption”. Also change sorption by adsorption along the manuscript.
Major revisions
- Abstract: Important conclusions have not been included in abstract. Avoid not specific conclusions such as “It was observed that the increased concentration of acid accelerates the catalytic activity”. Really the acid concentration does not accelerate the catalytic activity, the acid concentration affects the textural and surface chemistry of carbon and, this is the reasons why the catalytic activity is affected.
- Line 66. Ce2+?. The most common oxidation states of cerium are 3+ and 4+.
- Isotherms must be properly described. As the authors indicates in line 88-89, samples show a sharp increase in adsorption at low relative pressures which indicates a considerable volume of micropores. Thus, the isotherms are not a pure Type IV, but a mixture of Type I-Type IV isotherms characteristic of micro-mesoporous materials.
- SEM images reveals the presence of macroporosity. Thus, to properly characterize activated carbons and how the functionalization and cerium deposition can affect or block such porosity, Hg-porosimetry is required.
- Line 100-101. The authors state that “The enhancement of the specific surface area can be explained by the formation of new oxygen-containing functional groups on the surface of activated carbon”. The used reference does not justify this fact. Those authors ascribed the enhancement of surface area after the HNO3 treatment to the removal of the impurities attached on the surface including metal ion. Really, a porosity blockage would be expected by the introduction of new oxygenated surface groups (see, carbon 68 (2014) 520-539).
- Inorganic matter of activated carbons must be determined (e.g. by TGA in air) to see the influence of nitric acid to the impurity removal and thus, the porosity enhancement.
- How do the authors explain this fact: line 114-115 “the higher concentration of the oxidizing agent, the narrower micropores will be formed?
- Line 118-124. The authors indicate that: “The introduction of Ce resulted in the increase of the specific surface area and the volume of micropores, regardless of the applied concentration of the oxidizing agent”… “Nevertheless, the reference research was not focused on activated carbon used as a support for Ce. What is more, the obtained results are confirmed by SEM images, suggesting that the promoting effect of ceria on the development of specific surface area may occur only for specific supports”. However, SEM images does not clearly point out this statement. Usually, the introduction of ceria (or metals) by incipient wetness impregnation to carbon materials, could cause a partial porosity blockage. It is true that the ceria content is not very high, but an increase of surface is not expected by metal deposition. Nevertheless, ceria is a well-known combustion catalyst, which is able to transfer active oxygen from the ceria to the carbon surface and thus, to produce some carbon gasification. Could a temperature of 250ºC enough to produce such gasification and thus, could this explain the surface area enhancement observed after the deposition of cerium on the carbon surface?. Could the ceria catalyse the decomposition of N or O functional groups opening in some way the porosity of carbon?. AC/ceria reference catalyst can be useful to analyse all these effects.
- Line 140-141. The authors state that “The diffraction maxima of Ce deposited on the surface of AC were not detected during the experiment, probably due to its very small amount that is well-dispersed on the support”. This is true but, according to the authors, if the white particles observed in Figure 8 are attributed to CeO2, the average mean particle seem to be around 0.35-0.45 mm, thus, diffraction peaks must be observed. Elemental mapping in Figure 8 can be useful to really identify Cerium in the SEM image.
- Ceria dispersion must be studied by HRTEM.
- Cerium nitrate decomposed between 190-300 ºC depending on the conditions, is the decomposition treatment of cerium nitrate (250 ºC for 1 h) enough to decompose the precursor salt to form CeO2?.
- It would be important to known the surface chemistry of samples before to analyse the TGA data. Therefore, I consider that elemental analyses must be discussed before TGA results.
- Line 187-188. Authors indicates that “there is a noticeable difference between the weight loss of the samples non-doped and doped with Ce during the second stage of decomposition, and it progressively increases with the increasing temperature”. Could be this fact explained based on the character of ceria as combustion catalysts.
- The stability of the catalysts must be also analysed. For example, several reused cycles must be performed or stability test under long reaction times.
- Scale on Figure 7 c and d is required. Nevertheless, it is not correct to state that “In Fig 7c and 7d taken at higher magnification (20 000), the pores of the samples are visible. The pores are not blocked by CeO2, thus its small particles are well-dispersed on the surface of activated carbon”. The microporosity and narrow mesoporosity of sample (around 2-3 nm, according to Table 1) is not possible to be detected by SEM. The same occurs with the following statement obtained from SEM images: “Since the amount of Ce introduced onto the surface of activated carbon was very small, adherence of CeO2 particles on the support did not lead to the blockage of pores”.
- EDX analysis. The authors indicate that “the amount of the deposited CeO2 was dependent on the concentration of the oxidizing agent. Hence, it is possible that more violent interaction between concentrated acid and activated carbon decreased the capacity of AC surface to adsorb cerium oxide”. Additionally, after introduction of Ce, the impurities of Si traces disappeared”. This statement is doubtful, because if the cerium is introduced by incipient wetness impregnation of the carbon support, all the cerium precursor is deposited on the carbon surface and after decomposition, similar amount of ceria on the carbon support could be expected independently of the nitric acid concentration used. Other thing would be how the oxygenizing functionalization process affect to the dispersion of such ceria, but the quantity would be similar. Other question that I can understand is why the amount of Si decreases after the cerium introduction.
- XPS and ceria dispersion analyses are required for a proper discussion of the catalytic activity in order to know the Ce chemical nature and active sites availability for NO reduction.
- There is a lot of Figure in the catalytic section. Try to combined some of them. For example, similar conclusions can be obtained from Figures 9 and 12.
- Line 297-298. The authors ascribed the highest activity of 14U to the abundance of functional N-containing groups introduced onto the surface of AC. However, the N content is 41.00, 41.31 and 42.35 % for 5U, 10 U, 14U. This N quantity is very similar, at least for U5 and 10 U samples, whereas the activity is very different (T50). Thus, the N content does not explain this activity differences.
Line 307-308. Here, the highest activity of 14U is explained based on its higher concentration of active oxygen. Please unify this explanation.
- The effect of N—groups was discussed based on bibliographic results: “The results indicated that pyridine, pyrrole and quaternary amine groups were the centres for NO adsorption while the phenolic hydroxyl group was an adsorption site for NH3”. Based on this, the authors concluded that “Thus, it is predicted that in case of the analysed samples, pre-treatment with the most concentrated acid introduced the highest amount of these groups”, but this is a very speculative conclusion. XPS analysis could be useful to characterize the surface chemistry of samples and to correlate these properties with the different catalytic activity.

Author Response
The enhanced performance of N-modified activated carbon promoted with Ce in selective catalytic reduction of NOx with NH3
In this manuscript, N and O functionalized activated carbons were prepared by treatment with urea and nitric acid. Moreover, ceria was also introduced into the functionalized activated carbon by incipient wetness impregnation. The catalytic performance of the synthesized activated carbons was tested in the NO reduction in presence of NH3. The samples have been extensively characterized; however, the results are not properly discussed. Thus, I consider that the quality of the manuscript is not enough for its publication in Catalysts and several issues must be improved previous to its further consideration for publication:
Minor revisions:
- English must be checked and corrected. There are some sentences in which the verb is missing, e.g.: abstract: “the goal of the study to modify activated carbon with …”. Pag 1. Line 27: isadvantages must be disadvantages.
Response: Thank you for your valuable remarks about our minor spelling errors. We revised once again the whole manuscript and corrected all of the language, spelling and grammar mistakes that we had made.
2. Table 1. The equations and methods used to calculate all the parameters of Table 1 must be defined. Is “Pore volume” the volume of micropores? Is total pore volume calculated as the N2 adsorbed at 0.95 minus the micropores volume? Has the average pore diameter been calculated from the DFT pore size distribution?
Response: Thank you for the remark. Yes, the column “pore volume” was dedicated to the volume of micropores, unfortunately we omitted the name “micropores”, which was corrected in the actual version of the manuscript.
The average pore diameter was calculated using BJH pore size distribution method, using Kelvin-Thomson equation. Total pore volume was obtained from the pore volume distribution curve. It was calculated by the differentiation of the function V = f(rk) where V is pore volume while rk is a radius of the pore derived from Kelvin equation.
- For a proper characterization of activated carbons, Hg-porosimetry must be also performed in order to characterize the wide mesoporosity and macroporosity.
Response: Thank you very much for your valuable remark. Unfortunately, due to very short time for the revision, we are not able to perform Hg-porosimetry experiment. However, in our future work we will consider to extend our analysis with this kind of measurements, which will be an added value to the research.
4. Line 98. “Total pore volume from 0.36 cm3 g-1 to 0.50 cm3 g-1”. The total pore volume increases from 0.45 to 0.50 cm3 g-1 when the activated carbon is treated with 5 M acid.
Response: Of course, you are absolutely right, thank you very much. That was a big oversight from our site and we corrected it in the manuscript.
5. Line 129-130. “The presence of those maxima confirms the existence of graphene sheets and disordered nature of activated carbon 36”. Is this 36 the reference?
Response: Thank you for the remark, yest, you are right – this is a reference. Due to small oversight, we missed the squared brackets. Nevertheless, right now it is corrected in the manuscript.
6. Figure 7. Figure caption (b) is missing.
Response: Thank you for the remark. We changed slightly the presentation of SEM images in the manuscript. Since then, right now all of the captions are included in the figures.
7. Line 309. Correct “non-modifiedactivated carbon”
Response: Thank you for the remark, however, we changed slightly that part of the manuscript. Thus, both the mistake and the entire sentence were removed from the paragraph.
8. Equation 6 is not correct. Remove the number 2 of ceria to properly balance the equation
Response: Thank you, we corrected the equation.
9. Which activated carbon was used? Is it a commercial one or was it synthesized in your laboratory?
Response: The activated carbon that was utilized for the purpose of our studies was provided by Gryfskand (Hajnówka), thus it was a commercial one. We are sorry for our oversight and not including that information in the “Catalyst preparation” section earlier. Right now the information appears in the paragraph devoted to the materials and methods used within the research.
10. Table 2. This table does not reflect the real preparation conditions because samples with U (5U, 10 U and 14 U) were also modified with urea, whereas the samples with Ce were treated with the same procedure described for U samples but Ce was also introduced. Please modify this table in order to avoid misunderstanding.
Response: We introduced additional columns to Table 2. in order to avoid misunderstanding. According to your suggestion, we clarified the issue related to the modifications with urea and Ce. We hope that right now everything is listed in the table in more direct manner.
11. It is better use “nitrogen adsorption at -196 ºC” rather than “low-temperature nitrogen sorption”. Also change sorption by adsorption along the manuscript.
Response: According to your suggestion, we changed the word “sorption” into “adsorption” along the manuscript and replaced “low-temperature nitrogen sorption” by “nitrogen adsorption at -196 ºC” which, indeed, is much more accurate.
Major revisions
- Abstract: Important conclusions have not been included in abstract. Avoid not specific conclusions such as “It was observed that the increased concentration of acid accelerates the catalytic activity”. Really the acid concentration does not accelerate the catalytic activity, the acid concentration affects the textural and surface chemistry of carbon and, this is the reasons why the catalytic activity is affected.
Response: Thank you for the comment, according to which we corrected our abstract and introduced specific conclusions that we made basing on the obtained results. We hope that right now the assumptions presented in abstract according to the influence of acid on the catalytic activity are much more accurate in comparison to the previous ones.
2. Line 66. Ce2+?. The most common oxidation states of cerium are 3+ and 4+.
Response: We are very sorry for that mistake that we made. Thank you very much for the remark, it was corrected in the manuscript.
3. Isotherms must be properly described. As the authors indicates in line 88-89, samples show a sharp increase in adsorption at low relative pressures which indicates a considerable volume of micropores. Thus, the isotherms are not a pure Type IV, but a mixture of Type I-Type IV isotherms characteristic of micro-mesoporous materials.
Response: Thank you for the comment. We corrected the description of nitrogen adsorption at -196ºC, we included appropriate citation and emphasized that the obtained isotherms are a combination of I and IV type, according to IUPAC classification.
4. SEM images reveals the presence of macroporosity. Thus, to properly characterize activated carbons and how the functionalization and cerium deposition can affect or block such porosity, Hg-porosimetry is required.
Response: Thank you very much for your valuable remark. Unfortunately, we are not able to perform Hg-porosimetry experiment to determine how the functionalization and deposition of Ce influenced the texture of the analyzed activated carbon. However, in our future work, surely, we will include Hg-porosimetry measurement in the characterization procedure.
5. Line 100-101. The authors state that “The enhancement of the specific surface area can be explained by the formation of new oxygen-containing functional groups on the surface of activated carbon”. The used reference does not justify this fact. Those authors ascribed the enhancement of surface area after the HNO3 treatment to the removal of the impurities attached on the surface including metal ion. Really, a porosity blockage would be expected by the introduction of new oxygenated surface groups (see, carbon 68 (2014) 520-539).
Response: Thank you very much for the comment. According to your valuable suggestion, we supported our conclusions by citing more accurate literature (including the one that you recommended).
6. Inorganic matter of activated carbons must be determined (e.g. by TGA in air) to see the influence of nitric acid to the impurity removal and thus, the porosity enhancement.
Response: Thank you very much for the valuable suggestion, nonetheless we are not able to carry out TGA measurement in air in such a limited time of revision.
7. How do the authors explain this fact: line 114-115 “the higher concentration of the oxidizing agent, the narrower micropores will be formed?
Response: Thank you for the comment. We changed the word “will” into “are” formed, due to the fact that it is the result obtained within the research carried out by us. We predict that, possibly, the entrance to the pores was partly covered by the introduced N-containing groups which caused that the pores become narrower.
8. Line 118-124. The authors indicate that: “The introduction of Ce resulted in the increase of the specific surface area and the volume of micropores, regardless of the applied concentration of the oxidizing agent”… “Nevertheless, the reference research was not focused on activated carbon used as a support for Ce. What is more, the obtained results are confirmed by SEM images, suggesting that the promoting effect of ceria on the development of specific surface area may occur only for specific supports”. However, SEM images does not clearly point out this statement. Usually, the introduction of ceria (or metals) by incipient wetness impregnation to carbon materials, could cause a partial porosity blockage. It is true that the ceria content is not very high, but an increase of surface is not expected by metal deposition. Nevertheless, ceria is a well-known combustion catalyst, which is able to transfer active oxygen from the ceria to the carbon surface and thus, to produce some carbon gasification. Could a temperature of 250ºC enough to produce such gasification and thus, could this explain the surface area enhancement observed after the deposition of cerium on the carbon surface?. Could the ceria catalyse the decomposition of N or O functional groups opening in some way the porosity of carbon?. AC/ceria reference catalyst can be useful to analyze all these effects.
Response: Thank you very much for the comment. According to your suggestion, we decided to withdraw the statement that SEM images are a proper confirmation of the development of the specific surface area of activated carbon after modification with ceria, since it is not a direct confirmation. In the part of the manuscript that you cited, we emphasized that our results are not in agreement with the literature of the subject: “In fact, the result is not in agreement with most of the studies over Ce-doped catalysts that indicate the decrease in SBET after introduction of less than 1 wt.% of ceria [7–9].” Due to the fact that you are absolutely right that the introduction of metals by incipient wetness impregnation into carbon materials can cause a porosity blockage, we tried to do our best to find accurate citations that explain the results that we obtained, according to your suggestions.
9. Line 140-141. The authors state that “The diffraction maxima of Ce deposited on the surface of AC were not detected during the experiment, probably due to its very small amount that is well-dispersed on the support”. This is true but, according to the authors, if the white particles observed in Figure 8 are attributed to CeO2, the average mean particle seem to be around 0.35-0.45 mm, thus, diffraction peaks must be observed. Elemental mapping in Figure 8 can be useful to really identify Cerium in the SEM image.
Response: Thank you very much for making a very good point concerning this issue.
No ceria was detected by XRD, due either to its low content and/or non-crystalline structure We corrected the description of XRD as follow “No ceria was detected by XRD, due either to its low content and/or non-crystalline structure. The letter trendy partly back up by the fact on SEM images in some places 0.35 - 045 nm particles are visible. Thus it not exclude however the additional well dispersed active phase in porous.
10. Ceria dispersion must be studied by HRTEM.
Response: Thank you for the suggestion, nevertheless, due to the limited time predicted for the corrections of our manuscript, unfortunately, we are not able to perform HRTEM studies.
11. Cerium nitrate decomposed between 190-300 ºC depending on the conditions, is the decomposition treatment of cerium nitrate (250 ºC for 1 h) enough to decompose the precursor salt to form CeO2?.
Response: Thank you for the question and comment. According to the scientific literature (J. Inorg. Nucl. Chem., 1961, vol. 17, pp. 281 to 285), total decomposition of cerium nitrate hydrate occurs at around 265ºC when the catalyst is calcined for 64 min. Since we deposited a small amount of cerium nitrate hydrate on the surface of the support, we decided to apply slightly lower temperature of the procedure and somewhat longer time of the procedure. Because any nitrate groups were detected by FT-IR after calcination of Ce-containing catalysts, we can conclude the temperature and duration of the calcination procedure was appropriately chosen.
12. It would be important to known the surface chemistry of samples before to analyse the TGA data. Therefore, I consider that elemental analyses must be discussed before TGA results.
Response: Thank you for the suggestion, we moved EDX analysis before TGA results discussion.
13. Line 187-188. Authors indicates that “there is a noticeable difference between the weight loss of the samples non-doped and doped with Ce during the second stage of decomposition, and it progressively increases with the increasing temperature”. Could be this fact explained based on the character of ceria as combustion catalysts.
Response: Thank you very much for the comment, according to which we refilled the explanation of the weight loss of the samples doped with Ce:
“Additionally, ceria is recognized as an active combustion catalyst and that led to the partial combustion of activated carbon at temperatures higher than 380 oC”.
14. The stability of the catalysts must be also analysed. For example, several reused cycles must be performed or stability test under long reaction times.
Response: Thank you very much for the suggestion. We will consider the analysis of the stability of the catalyst in our further studies due to the fact that undoubtedly this kind of experiments will give us a better view of the usefulness of the catalysts in industrial applications. Nevertheless, considering very limited time expected for the improving the text, we are not able meet your expectations on that point and perform the stability tests on the samples.
15. Scale on Figure 7 c and d is required. Nevertheless, it is not correct to state that “In Fig 7c and 7d taken at higher magnification (20 000), the pores of the samples are visible. The pores are not blocked by CeO2, thus its small particles are well-dispersed on the surface of activated carbon”. The microporosity and narrow mesoporosity of sample (around 2-3 nm, according to Table 1) is not possible to be detected by SEM. The same occurs with the following statement obtained from SEM images: “Since the amount of Ce introduced onto the surface of activated carbon was very small, adherence of CeO2 particles on the support did not lead to the blockage of pores”.
Response: Thank you for your recommendations and comments about SEM images. We slightly changed the presentation of SEM results and we decided to compare the sample 14UCe at two magnifications (scale is included on both images). We also corrected the statement about the detection of porosity by SEM.
16. EDX analysis. The authors indicate that “the amount of the deposited CeO2 was dependent on the concentration of the oxidizing agent. Hence, it is possible that more violent interaction between concentrated acid and activated carbon decreased the capacity of AC surface to adsorb cerium oxide”. Additionally, after introduction of Ce, the impurities of Si traces disappeared”. This statement is doubtful, because if the cerium is introduced by incipient wetness impregnation of the carbon support, all the cerium precursor is deposited on the carbon surface and after decomposition, similar amount of ceria on the carbon support could be expected independently of the nitric acid concentration used. Other thing would be how the oxygenizing functionalization process affect to the dispersion of such ceria, but the quantity would be similar. Other question that I can understand is why the amount of Si decreases after the cerium introduction.
Response: Thank you for the comments. According to the issue of the amount of Ce – you are right, it is true that acid treatment was not the reason of higher amount of Ce and we explained it in the manuscript.; Secondly, indeed, concerning the presence of silica or silica species in activated carbon it is an impurity connected to ash that is present in the used activated carbon (this was a commercial sample). Apart from sorption on activated carbon sample there could have been sorption of cerium on a Si-containing ash.
By the way, this is quite an interesting idea and we intend to investigate it in our future work. We included the additional explanation of this issue in the manuscript.
17. XPS and ceria dispersion analyses are required for a proper discussion of the catalytic activity in order to know the Ce chemical nature and active sites availability for NO reduction.
Response: XPS would undoubtedly be an added value to our manuscript. Nevertheless, we are very sorry to admit that we are not able to perform XPS analysis in such a limited time of 7 days.
18. There is a lot of Figure in the catalytic section. Try to combined some of them. For example, similar conclusions can be obtained from Figures 9 and 12.
Response: Thank you for the remark. According to your suggestion, we combined the figures, so that the results are more clear and visible.
19. Line 297-298. The authors ascribed the highest activity of 14U to the abundance of functional N-containing groups introduced onto the surface of AC. However, the N content is 41.00, 41.31 and 42.35 % for 5U, 10 U, 14U. This N quantity is very similar, at least for U5 and 10 U samples, whereas the activity is very different (T50). Thus, the N content does not explain this activity differences.
20. Line 307-308. Here, the highest activity of 14U is explained based on its higher concentration of active oxygen. Please unify this explanation.
Response for comments 19 and 20: Thank you very much for the remark. The matter of oxidation is complicated by the fact that both the initial structure of carbonaceous materials and the type of oxidative pretreatment play an important role and only semi-quantitative prediction on the type and number of groups may be generally given. The most general conclusion is that the type of surface groups, and, as a consequence acidic/basic character of the surface depend predominantly on the type of oxidant. The number of the groups depends additionally on the concentration of the oxidant, temperature of the treatment and its duration. The influence on NO conversion is more complex, not only N species are responsible for the catalytic activity, but also O species. We unified the explanation in the manuscript.
21. The effect of N—groups was discussed based on bibliographic results: “The results indicated that pyridine, pyrrole and quaternary amine groups were the centres for NO adsorption while the phenolic hydroxyl group was an adsorption site for NH3”. Based on this, the authors concluded that “Thus, it is predicted that in case of the analysed samples, pre-treatment with the most concentrated acid introduced the highest amount of these groups”, but this is a very speculative conclusion. XPS analysis could be useful to characterize the surface chemistry of samples and to correlate these properties with the different catalytic activity.
Response: Thank you very much for the valuable comment. However, we are not able to perform XPS studies in such a short time. However, you are right that it sounds speculative, so perhaps one of possibilities is that in case of in case of the analysed samples, pre-treatment with the most concentrated acid may have introduced the highest amount of these groups. Another possibility are either the different distribution of such groups.

Round 2
Reviewer 1 Report
The changes made by the authors in relation to the comments are satisfactory to recomend publication.
Reviewer 2 Report
The authors have attended to the most of my comments. I consider this work can be published in Catalysts at its current state.